# Si-GT: Fast Interconnect Signal Integrity Analysis for Integrated Circuit Design via Graph Transformers

**Yuting Hu**[1] **Tarek Mohamed**[2] **Chenhui Xu**[1] **Hua Xiang**[3] **Hussam Amrouch**[2]
**Gi-Joon Nam**[3] **Jinjun Xiong**[1]*

[1]University at Buffalo, Buffalo, NY, USA
[2]University of Stuttgart, Stuttgart, Germany
[3]IBM Research, Yorktown Heights, NY, USA

## Abstract

Signal integrity issues present significant challenges in modern integrated circuit (IC) design, as crosstalk-induced delay variation and transient glitches caused by capacitive coupling among interconnects can severely impact IC functional correctness. Although circuit simulators like SPICE can deliver accurate signal integrity analysis, their computational cost becomes prohibitive for large-scale designs. In this paper, we propose Si-GT, a novel transformer-based model for fast and accurate signal integrity analysis in IC interconnects. Our model elaborates three key designs: (1) virtual NET token to encode net-specific signal characteristics and serve as net-wise representation, (2) mesh pattern encoding to embed high-order mesh structures at each node while distinguishing uncoupled wire segments, and (3) intra-inter net (IIN) attention mechanism to capture structures of signal propagation path and coupling connections. To support model training and evaluation, we construct the first interconnect signal integrity dataset comprising 200k delay examples and 187k glitch examples using SPICE simulations as the golden reference. Our experiments show that our Si-GT surpasses state-of-the-art graph neural network and graph transformer baselines with substantially reduced computation compared to SPICE, offering a scalable and effective solution for interconnect signal integrity analysis in IC design verification. We release the code, model, and datasets at https://github.com/xlab-ub/Si-GT.

## 1 Introduction

Signal integrity (SI) analysis is essential in integrated circuit (IC) design to ensure reliable signal transmission and correct timing behavior (Caignet et al., 2001). Among signal integrity problems, crosstalk is the primary culprit. Dense interconnect layouts and high-speed signaling in modern ICs exacerbate crosstalk-induced noise and delay variations, leading to potential functional errors, performance degradation, and even chip failure Li et al. (2022); Song et al. (2015). Engineers have to run SPICE simulations (Quarles et al., 1994) repeatedly throughout IC design flow to identify circuit behavior and crosstalk-induced noise and delay violations, allowing careful crosstalk mitigations (Vittal & Marek-Sadowska, 1997; Stöhr et al., 1998; Duan et al., 2010; Gao & Liu, 1996), which is computationally prohibitive for very-large-scale integration (VLSI) (Achar & Nakhla, 2001).

Recently, machine learning (ML) has emerged as a computationally efficient surrogate for signal integrity analysis in IC design (Kahng et al., 2015; Lu & Lim, 2022; Swaminathan et al., 2020; Wang & Luo, 2019; Cheng et al., 2020; Liang et al., 2022; Liu et al., 2025). However, most prior efforts primarily concentrate on timing prediction, aiming to "unravel the mystery" of black-box timing estimation formulas in sign-off timers. These works generally do not model crosstalk effects explicitly with aggressor–victim switching interactions and signal pattern-dependent analysis.

---

*Corresponding author: `jinjun@buffalo.edu`

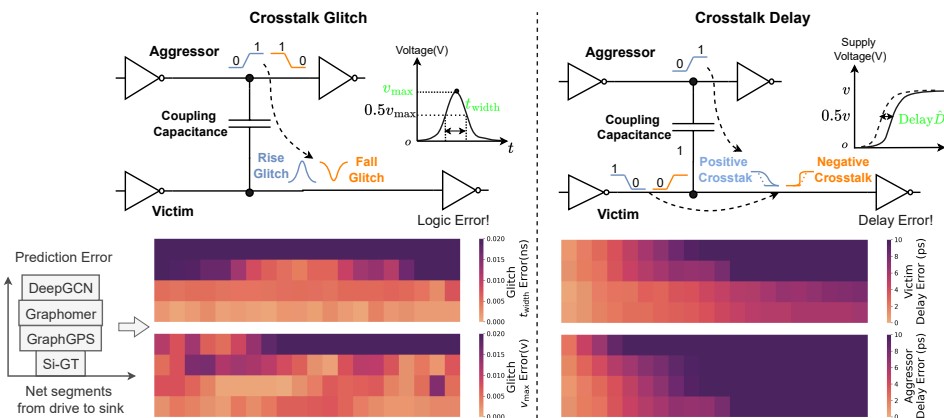

Figure 1: Prediction tasks illustration and Si-GT model performance.

Advances in graph neural networks (GNNs) (Wu et al., 2020) and graph transformer (GT) (Dwivedi & Bresson, 2020) have revolutionized machine learning capabilities for graph-structured data, enabling breakthrough applications in electronic design automation (EDA) from precise timing (Guo et al., 2022; Hu et al., 2023; Lin et al., 2025; Zhong et al., 2024; Guo et al., 2025) and parasitics prediction (Ren et al., 2020; Shahane et al., 2023; Yoon et al., 2025; Liu et al., 2023) to complex optimization tasks like placement (Lu et al., 2020; Ding et al., 2024; Hou et al., 2025) and routing (Cheng & Yan, 2021; Liao et al., 2020; Wang et al., 2024). However, developing a graph learning model for signal integrity analysis is challenging due to the complex crosstalk effect (Aragones & Rubio, 2003). In IC interconnects, the crosstalk effect arises from electromagnetic interference between signals propagating on adjacent wires. On the one hand, the severity and nature of this interference depend on multiple factors, including switching directions, active/quiet net states, slew rate, coupling capacitance, and wire characteristics (Wong et al., 2000; You & Soma, 1990). On the other hand, the crosstalk effect exhibits both long-range dependencies (i.e., signal propagating from drive to distant load) and adjacent net-wise dependencies (i.e, energy transfer between coupled nets). The successful application of GNNs to EDA tasks relies on incorporating domain physics into the graph's inductive bias (Haoxiang et al., 2022). To design an effective graph learning model for signal integrity analysis, it's important to encode both signal switching patterns and structural features into the graph inductive bias while accounting for the unique circuit behaviors under crosstalk effect.

Graph transformers are excellent at capturing long-range dependencies through self-attention mechanisms. To this end, we propose Si-GT, a novel graph transformer model for IC interconnect signal integrity analysis. Si-GT incorporates three key designs: (1) Mesh pattern encoding, which embeds local mesh structures at each node to enrich node features and separate uncoupled nets; (2) Virtual <NET> tokens, which encode net-specific signal characteristics (e.g., switching direction and slew rate) and serve as net-level representations, with their receptive fields restricted to the corresponding nets via attention masks; (3) Intra–Inter Net (IIN) attention, which explicitly models both the spatial relationships among nodes within a net and the coupling effects from adjacent nets connected by coupling capacitors. Our contributions are summarized as follows:

- We propose Si-GT, a Transformer-based model for fast interconnect signal integrity analysis. To enhance graph inductive bias, Si-GT leverages virtual NET tokens for net-level signal encoding, mesh pattern encoding for local coupling structures, and intra-inter net attention to capture signal propagation and coupling effects.

- We construct a dataset for ML-based signal integrity analysis of IC circuits, comprising 200,200 crosstalk delay examples and 187,309 crosstalk glitch examples referring to golden SPICE simulations. To the best of our knowledge, this is the first large-scale dataset dedicated to IC interconnect signal integrity analysis.

- Experiments highlight the superior performance of Si-GT over advanced GNNs and graph transformers, as well as in computational efficiency compared to SPICE simulation. We validate the effectiveness of each design in Si-GT through ablation studies.

## 2    RELATED WORK

**Crosstalk Effect.** Crosstalk is a severe signal interference that degrades signal integrity in circuits (Hall & Heck, 2011). As illustrated in Figure 1, when a signal transitions on the interconnect (aggressor), it induces a voltage disturbance on the adjacent interconnect (victim). This interference can manifest either as a crosstalk glitch on the victim, leading to a logic error, or as crosstalk-induced delays in signal propagation, causing timing failures (Vittal et al., 1999). Different switching patterns of aggressor and victim can create distinct delay scenarios. When aggressor and victim switch in the same direction, constructive interference occurs, accelerating the victim's transition and potentially causing timing violations. When they switch in opposite directions, destructive interference occurs, slowing the victim's transition and increasing delay (Wong et al., 2000; You & Soma, 1990).

**ML for SI.** ML has been applied to reduce the cost of SI analysis in circuit design cycles (Lu & Lim, 2022). Related studies mainly fall into three categories: (1) early-stage crosstalk mitigation at the global routing stage, including critical net classification (Liang et al., 2020; 2022), crosstalk-aware placement (Gao et al., 2022; Yu et al., 2025), gate sizing (Zhou et al., 2022; Lu et al., 2021), and buffer insertion (Ding et al., 2024); (2) pre-routing timing estimation (Jin et al., 2024); (3) post-routing timing estimation (Kahng et al., 2015; Cheng et al., 2020; Liu et al., 2025; Ye et al., 2023). These works for SI only serve for timing prediction and share a key limitation that none of them consider signal pattern variability in both their dataset and model design, which is central to accurate and practical signal integrity analysis.

**Graph Transformer.** Graph transformer (Kreuzer et al., 2021; Yuan et al., 2025; Ying et al., 2021) encodes structural information into the graph inductive bias and leverages the graph attention mechanism to capture the long-range dependencies, breaking the limitation of message-passing GNNs in capturing global context due to its inherent over-smoothing and over-squashing issues (Pei et al.). A graph transformer layer is composed of a self-attention module followed by a feed-forward neural network (FFN). Given a graph $\mathcal{G}$ having $n$ nodes with node feature matrix $X \in \mathrm{R}^{n \times d}$ where $d$ is node feature dimension, self-attention module will project $X$ into query, key, and value matrices: $Q = XW_Q$, $K = XW_K$, and $V = XW_V$ with three trainable weight matrices $W_Q, W_K \in \mathrm{R}^{d \times d_K}$, $W_V \in \mathrm{R}^{d \times d_V}$ respectively. Then global attention is calculated with self-attention module: $\mathrm{Attn}(X) = \mathrm{softmax}(\frac{QK^T}{\sqrt{d_K}})V$.

## 3    FORMULATION AND BACKGROUND

### 3.1    INTERCONNECT FEATURIZATION

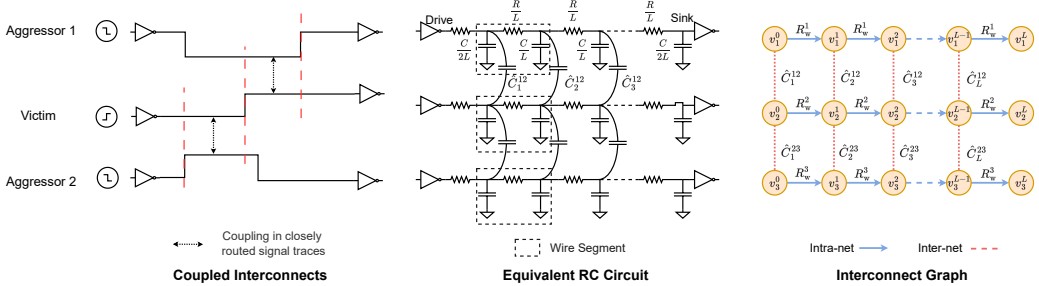

Figure 2: Graph representation for coupled IC interconnects.

IC interconnects are typically modeled as distributed RC circuits using wire-load models (Jin et al., 1999). As shown in Figure 2, for three coupled interconnects (two aggressors and one victim) driven and loaded by inverters, its equivalent RC circuit can be derived by breaking the wire into $L$ equal-length segments with $\Pi$ model (Chu & Wong, 2001) and parameterized with wire parasitics extracted from the physical layout. For each interconnect $net^i$, we define the wire resistance as $R_{\mathrm{w}}^i$ and wire capacitance as $C_{\mathrm{w}}^i$ for every segment, while $\hat{C}_s^{ij}$ denotes the coupling capacitance between $net^i$ and $net^j$ at segment $s$.

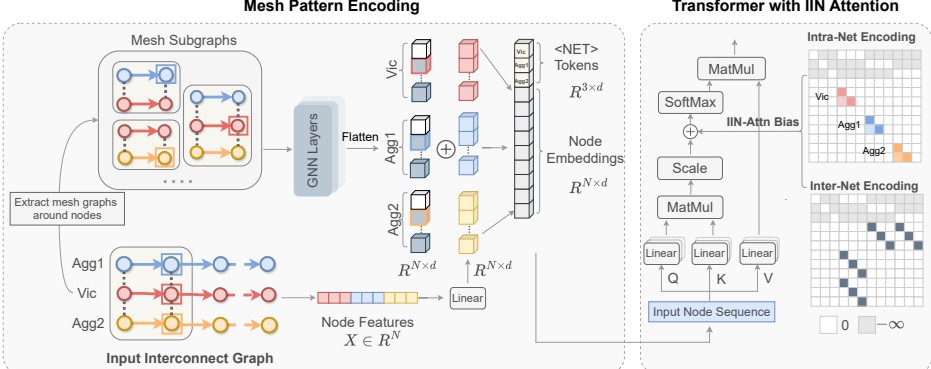

Figure 3: Overview of Si-GT.

Given a configuration of $M$ coupled interconnects $\left\{net^i\right\}_{i=1}^M$, as shown in Figure 2, we represent its equivalent RC circuit as a graph $\mathcal{G}(\mathcal{V}, \mathcal{E})$. Here, $\mathcal{V}$ denotes the vertex set and $\mathcal{E}$ denotes the edge set. Each net $net^i$ is a node subset $\mathcal{V}_S^i = \left\{v_i^0, v_i^1, v_i^2, ..., v_i^L\right\} \subset \mathcal{V}$, where $L$ is the number of length-equivalent wire segments. Nodes in $\mathcal{V}_S^i$ are connected in their physical sequence along $net^i$ and characterized by the wire capacitance $C_w$, and each edge is associated with the wire resistance $R_w^i$. Edges in the graph represent two types of connections: intra-net connections between nodes within the individual net, and inter-net connections between nodes of different nets connected with coupling capacitors. For intra-net connections, we assign $[R_w, 0]$ as the edge feature of wire connection, while for inter-net connections, we assign $[0, \hat{C}]$ as the edge feature of coupling connection. More graph construction methods are discussed in Appendix B.

## 3.2 PROBLEM DEFINITION

Crosstalk problems concerning the signal integrity in integrated circuits manifests through three fundamental scenarios: (1) victim net quiet; (2) victim net active and switching in the opposite direction to the aggressor; (3) victim net active and switching in the same direction to the aggressor. Given the above critical scenarios for signal integrity analysis, we address two prediction tasks:

**Task1: Crosstalk Glitch Prediction.** As shown in Figure 1, when the aggressor is switching and the victim is quiet, the potential difference across the coupling capacitor will generate a leakage current flowing to the victim, resulting an undesirable raising or falling glitch on the victim. The height and width of a glitch indicate the potential severity of a crosstalk event. Therefore, for the quiet victim case (1), we estimate two key parameters of crosstalk noise at each interconnect segment $s$ on the victim: the peak voltage $v_{\max}^s$ and noise width $t_{\text{width}}^s$. Here, noise width is defined as the time interval between the glitch's rising and falling edges at the 50% of its peak voltage.

**Task2: Crosstalk Delay Prediction.** As shown in Figure 1, when both aggressor and victim are switching, the coupling capacitor will affect the signal transition on victim , leading to slower or faster transition time by $\Delta t$. Net delay $D$ is defined as the time delay of a voltage waveform propagating through the net measured at the 50% of the waveform's voltage level. With crosstalk-induced variation, the net delay becomes $\hat{D} = D \pm \Delta t$. In scenarios (2) and (3) where both aggressor and victim nets are actively switching, we predict the net delay $\hat{D}_i^s$ of the signal propagating through each segment $s$ along the $net^i$.

## 4 SI-GT

In this section, we present Si-GT framework. As shown in Figure 3, Si-GT incorporates several key designs, including mesh pattern encoding, virtual NET tokens, and intra-inter net (IIN) attention. To integrate the structural information of interconnect graphs into the Transformer model and account for circuit-specific behaviors under crosstalk effects, we first decompose the interconnect graph into local mesh structures at each node and encode these structures using GNN layers for absolute positional encoding. Since the driving signals propagating along nets present different characteristics, we introduce virtual NET tokens to represent individual nets and encode the net-level information

such as net state, slew rate, and switching direction. Additionally, to structurally differentiate nodes within a specific net and coupled from different nets, we introduce IIN attention bias into the self-attention mechanism of Transformer to further improve the graph inductive bias.

## 4.1 MESH PATTERN ENCODING

In the equivalent RC circuit, coupling capacitors connect pairs of wire segments, forming a mesh unit defined as follows: [Couple Mesh Unit.] A couple mesh unit at a node $v_i^s$ on $net^i$ is defined as a subgraph of $\mathcal{G}(\mathcal{V}, \mathcal{E})$ with a set of nodes $\{v_i^{s-1}, v_i^s, v_j^{s-1}, v_j^s\}$ in direction from source to sink, assuming $net^i$ is coupled with an adjacent wire $net^j$ with coupling capacitance $\hat{C}_s^{ij}$. In coupled interconnects, we define mesh units to represent the coupling interactions between aligned segment pairs on aggressor and victim nets. As illustrated in Figure 3, for a node $v_i^s$ located at the end of the $s$-th segment on $net^i$, the number of mesh units constructed at $v_i^s$ is determined by the number of couplings with other nets. For example, if $net^i$ is coupled with two nets $net^j$ and $net^k$ at the $s$-th segment, we construct a subgraph $mesh(v_i^s)$ to capture its local structural information with two mesh units: $\{v_i^{s-1}, v_i^s, v_j^{s-1}, v_j^s\}$ and $\{v_i^{s-1}, v_i^s, v_k^{s-1}, v_k^s\}$. As shown in Figure 2, an aggressor net is typically coupled with a single victim net, while a victim net may be coupled with multiple aggressor nets. Therefore, we decompose the interconnect graph into local mesh subgraphs at each node, enabling the model to capture each node's local neighborhood while preserving the separation of uncoupled nets. Since mesh subgraphs are small, we then employ a shallow GNN model with $l$ layers $\text{GNN}^l$ to aggregate the local mesh structure information as the embedding of $v_i^s$, effectively encoding high-order mesh structural information into the node features. Finally, we add the $\text{GNN}^l$ embeddings to the linear projected node feature as the input to Transformer encoder:

$$h^{(0)}(v_i^s) = \text{GNN}^l(mesh(v_i^s)) + en(x(v_i^s)) \in \mathbb{R}^d \tag{1}$$

Since the interconnect is decomposed at the end node of each wire segment, we initialize the embeddings of driving nodes (i.e., the starting nodes of each net) to a zero vector: $h^{(0)}(v_i^0) = \mathbf{0} \in \mathbb{R}^d$.

## 4.2 INTRA-INTER NET ATTENTION MECHANSIM

Connections between nodes on each individual net are termed intra-net connections, while those linking a pair of coupled nets are referred to as inter-net connections. Both types play a critical role in the crosstalk effect. Intra-net connections capture incremental signal distortions and noise transformations along the net, whereas inter-net connections provide pathways for signal energy to transfer between nets. To capture this structural information, we introduce IIN-Attn, a novel attention mechanism that incorporates both intra-net and inter-net connections through specialized attention biases. First, as the signal is propagating forward on a single net, both the net delay and crosstalk noise attributes at any specific node are highly dependent on its former nodes that the signal has passed through. To this end, we design an intra-net encoding $\phi_{Intra}(v_i^u, v_i^v) : \mathcal{V} \times \mathcal{V} \to \mathbb{R}$ to capture the net structural feature and relative position between intra-nodes:

$$\phi_{\text{Intra}}(v_i^u, v_i^v) = \begin{cases} \dfrac{1}{d_{uv} \cdot R_{\text{w}}^i}, & \text{if } \{v_i^u, v_i^{u+1}, \dots, v_i^v\} \subseteq \mathcal{V}_S^i, \\ 0, & \text{otherwise.} \end{cases} \tag{2}$$

here, $d_{uv} = |v - u|$ denotes the distance from $v_i^u$ to $v_i^v$ along the net. $\phi_{Intra}(v_i^u, v_i^v)$ aggregates the wire resistance along the path from $u$ to $v$ on a net. If $u$ and $v$ are not from the same net, we set the value to be 0. The intra-net encoding explicitly captures the relative positional information of nodes connected in a net. Second, considering the interconnections between coupled nets, for $net^i$ corresponding to node set $\mathcal{V}_S^i$ and $net^j$ corresponding to node set $\mathcal{V}_S^j$, we define function $\phi_{Inter}(v_i^u, v_j^u) : \mathcal{V} \times \mathcal{V} \to \mathbb{R}$:

$$\phi_{\text{Inter}}(v_i^u, v_j^u) := \begin{cases} \hat{C}_{u+1}^{ij}, & \text{if net}^i, \text{net}^j \text{are coupled at } (u+1)\text{-th segment}, \\ 0, & \text{otherwise.} \end{cases} \tag{3}$$

$\phi_{Inter}(v_i^u, v_j^u)$ measures the coupling capacitance between $v_i^u$ and $v_j^u$ when the $(u+1)$-th net segment of $net^i$ and $net^j$ are coupled; otherwise, this value is set to 0. Intuitively, the inter-net encoding captures the connections between coupled net segments.

To encode structural information of coupled interconnects into attention layers, we directly incorporate the intra-net and inter-net biases into the attention logits:

$$\text{Attn-IIN}(X) = \text{softmax}\Big(\frac{QK^\top}{\sqrt{d_K}} + \tilde{\Phi}_{\text{IIN}} + \tilde{\Phi}_{\text{d}} + \tilde{\Phi}_{\text{sp}}\Big)V, \tag{4}$$

where the bias matrix $\tilde{\Phi}_{\text{IIN}}$ has entries given by: $\tilde{\phi}_{\text{IIN}} = \tilde{\phi}_{\text{Intra}} + \tilde{\phi}_{\text{Inter}}$. Here, $\tilde{\phi}_{\text{Intra}}$ and $\tilde{\phi}_{\text{Intra}}$ are obtained by applying learnable linear transformations to $\phi_{\text{Intra}}$ and $\phi_{\text{Intra}}$ respectively. Additionally, to capture global typological features, we use spatial encoding and edge encoding as extra attention bias terms to the attention module. Specifically, $\tilde{\Phi}_{\text{d}}$ encodes the distance of the shortest path (SP) between two connected nodes using learnable embedding table indexed by the distance scalar, while $\tilde{\Phi}_{\text{sp}}$ encodes the edge features along the path $\text{SP}_{ij} = (e_1, e_2, ..., e_n)$ from node $i$ to $j$ via $\phi_{\text{sp}}(i,j) = \frac{1}{n}\sum_{k=1}^{n} e_k w_k^T$. Here, $e$ is the edge feature, and $w \in \mathbb{R}^e$ is the weight embedding with edge feature dimension $\mathbb{R}^e$ (Ying et al., 2021).

### 4.3 Virtual NET Token

As signals propagate along interconnects from source to sink, local electromagnetic interference between adjacent wire segments not only affects signal integrity at individual segments but also accumulates, leading to significant distortions at the sinks of all nets. To capture this global net-level interaction, we introduce virtual <NET> tokens that represent individual nets and attend to all nodes in the self-attention mechanism. Besides, for net-level attributes such as the switching direction and slew rate of signals propagating on each net, Si-GT encodes these features into learnable embeddings of <NET> tokens. Specifically, for each distinct net, we assign a learnable embedding vector $h_{\text{<NET>}}^{(0)} \in \mathbb{R}^d$ as the input embedding for the special <NET> node. These embeddings are then processed alongside other node features within the transformer architecture. To restrict the receptive field of <NET> token to its corresponding net, as illustrated in Figure 3, we define an attention mask $\text{M}_{\text{NET}} \in \mathbb{R}^{|V| \times |V|}$ applied to the softmax logits of the IIN attention:

$$\text{M}_{\text{NET}}(i,j) := \begin{cases} -\infty, & \text{if } i \text{ represents } net^i \text{ and } j \notin \mathcal{V}_S^i, \\ 0, & \text{otherwise.} \end{cases} \tag{5}$$

$\text{M}_{\text{NET}}$ ensures that each <NET> node aggregates information exclusively from nodes within its respective net while remaining visible to all other nodes.

## 5 Experiments

### 5.1 Signal Integrity Dataset

To benchmark graph learning-based models for signal integrity analysis, we construct a dataset elaborating on crosstalk delay and glitch prediction tasks that covers various net lengths and signal characteristics. Our dataset is based on the circuits of two aggressors and one victim. To simulate the circuit behavior of coupled interconnects, we construct RC circuits of the interconnect wires for SPICE simulation. In practice, circuit simulation begins by converting a physical layout into an RC model through parasitic extraction, where wires are divided into multiple equal-length segments and replaced with RC networks. To model varying interconnect lengths, we sweep the number of segments and follow Intel's 14 FinFET (Fischer et al., 2015) to set the wire capacitance and resistance for every segment. Additionally, we sweep other key parameters such as wire separation, input slew rate, and signal switching direction to create diverse signal and coupling configurations, as summarized in Table 1. More details on coupling capacitance calculation, circuit simulation, and dataset construction pipeline are provided in the Appendix A.1. For each setup of RC circuit and its driving signals, we use Synopsys HSPICE simulator to measure the crosstalk delay and glitch along the nets, which results in 200,200 delay examples and 187,309 glitch examples in total.

### 5.2 Experimental Settings

**Baselines.** We compare our Si-GT model with following baselines: 1) GNNs, including standard GCN (Kipf & Welling, 2016), GAT (Veličković et al., 2017), GIN (Xu et al., 2018), GraphSAGE

Table 1: Circuit parameters of signal integrity dataset.

| Fixed Parameters | | | Sweeping Parameters | | | | | |
|---|---|---|---|---|---|---|---|---|
| Segment Length | Wire Resistance | Wire Capacitance | Net Length | Wire Separation | Coupling Capacitance | Victim State | Switching Direction | Slew Rate |
| 5 μm | 2.7 Ω/μm | 0.15fF/μm | 10-100 μm | 1-20 μm | 0.2214-7.908fF | Active/Quiet | Low-To-High/High-To-Low | 40-60ps |

(Hamilton et al., 2017), and advanced DeepGCN (Li et al., 2019) with residual connections; 2) recent SOTA graph transformers, including Graphomer (Ying et al., 2021), GraphGPS (Rampášek et al., 2022), and SGFormer (Wu et al., 2023); and 3) variations of Si-GT using different standard GNN backbones for mesh pattern encoding. We also evaluate Si-GT against baseline models using different position/ structural encodings (PE/SE), such as RWSE and LapPE (Dwivedi et al., 2021), as detailed in Appendix D.1.

**Settings.** In the main results, we use 5 convolutional layers for standard GNNs and 20 layers for DeepGCN. For Graphomer, we follow its original configuration with a 5-step limit for shortest path encoding, while GraphGPS is implemented with RWSE using 16 walk length. Full implementation details and experimental tests for hyperparameter setting of all baseline models are provided in Appendix C.3. For Si-GT, we use $l = 2$ GNN layers with a hidden dimension of 64 to encode the mesh patterns. We use 6 encoder layers with 4 attention heads and set the embedding size to 64 for the self-attention module and 128 for the feed-forward network. We train our Si-GT for 60 epochs with 256 batch size using the AdamW optimizer with polynomial learning rate decay and linear warmup, where the learning rate decays to 1e-9 over the total training steps, with weight decay set to 1e-4. All experiments in this paper are implemented with PyTorch 2.2.2, DGL 2.4.0, and Pytorch-geometric 2.6.1. Models are trained with $2 \times$ NVIDIA A100 80GB GPUs. Detailed training setup of baseline models are in Appendix C.1 due to space constraints.

## 5.3 EXPERIMENTAL RESULTS

Table 2: Mean relative accuracy (%) of crosstalk delay prediction results.

| Experiment | Metric | GNNs | | | | | Graph Transformer | | | | | | |
|---|---|---|---|---|---|---|---|---|---|---|---|---|---|
| | | GCN | GAT | GIN | SAGE | DeepGCN | SGFormer | Graphomer | GraphGPS | Si-GT GCN | Si-GT GAT | Si-GT GIN | Si-GT SAGE |
| AV Segment | $\hat{D}_{vic}$ | 65.21 | 58.68 | 60.02 | 58.50 | 85.49 | 64.64 | 88.23 | 88.23 | **88.32** | 88.28 | 88.27 | 88.28 |
| | $\hat{D}_{agg}$ | 57.38 | 43.27 | 54.17 | 62.41 | 71.64 | 52.60 | 72.58 | 72.65 | 73.67 | 73.18 | 73.30 | **73.81** |
| AV Sink | $\hat{D}_{vic}$ | 51.14 | 45.96 | 51.12 | 46.67 | 50.17 | 53.63 | 86.52 | 87.36 | 87.38 | **87.39** | 87.33 | 87.31 |
| | $\hat{D}_{agg}$ | 39.72 | 35.34 | 45.12 | 47.58 | 35.11 | 44.58 | 71.02 | 70.65 | 71.17 | **71.82** | 71.60 | 71.05 |
| V Segment | $\hat{D}_{vic}$ | 64.03 | 60.90 | 62.64 | 64.09 | 86.90 | 59.51 | 88.15 | 88.26 | 88.31 | 88.30 | 88.27 | **88.34** |
| V Sink | $\hat{D}_{vic}$ | 51.97 | 42.77 | 43.65 | 43.59 | 43.89 | 55.53 | 87.11 | 87.18 | 87.19 | 87.21 | **87.38** | 87.20 |

Table 3: Mean relative accuracy (%) of crosstalk glitch prediction results.

| Experiment | Metric | GNNs | | | | | Graph Transformer | | | | | | |
|---|---|---|---|---|---|---|---|---|---|---|---|---|---|
| | | GCN | GAT | GIN | SAGE | DeepGCN | SGFormer | Graphomer | GraphGPS | Si-GT GCN | Si-GT GAT | Si-GT GIN | Si-GT SAGE |
| V Segment | $t_{width}$ | 87.86 | 88.38 | 87.01 | 88.23 | 87.94 | 84.17 | 94.97 | 96.61 | 97.71 | 97.08 | **98.36** | 97.47 |
| | $v_{max}$ | 85.44 | 85.79 | 85.45 | 85.65 | 85.20 | 82.84 | 93.38 | **97.99** | 97.89 | 96.62 | 97.78 | 96.89 |
| V Sink | $t_{width}$ | 83.97 | 84.05 | 83.85 | 84.01 | 83.99 | 83.72 | 95.46 | 98.29 | **98.53** | 97.83 | 98.13 | 98.19 |
| | $v_{max}$ | 82.61 | 83.10 | 82.42 | 82.68 | 82.56 | 79.08 | 94.17 | 97.94 | **98.63** | 97.16 | 97.62 | 97.96 |

**Main Results.** We first report the main experimental results for crosstalk delay prediction in Table 2 and crosstalk glitch prediction in Table 3. Models are separately trained to predict delay and glitch metrics at each segment (Segment) along individual nets and specifically at their sinks (Sink). The sink-level results can provide an overview of model performance in predicting pin-to-pin delay and glitch. For delay prediction, we report accuracy for both aggressor and victim (AV) cases, as well as for victim-only (V) cases, since the victim is of greater concern in ensuring signal integrity. Predictions are evaluated against SPICE ground truth using mean relative accuracy.

The results show that: (1) Graph transformer models, particularly Graphomer, GraphGPS, and our Si-GT, consistently outperform traditional GNNs in signal integrity analysis, achieving significantly higher accuracy across both delay and glitch prediction tasks; and (2) for the more challenging delay prediction task, our proposed Si-GT model outperforms all baselines across all experiments. While GraphGPS also demonstrates strong performance, Si-GT variants consistently achieve the highest mean relative accuracy in nearly every case.

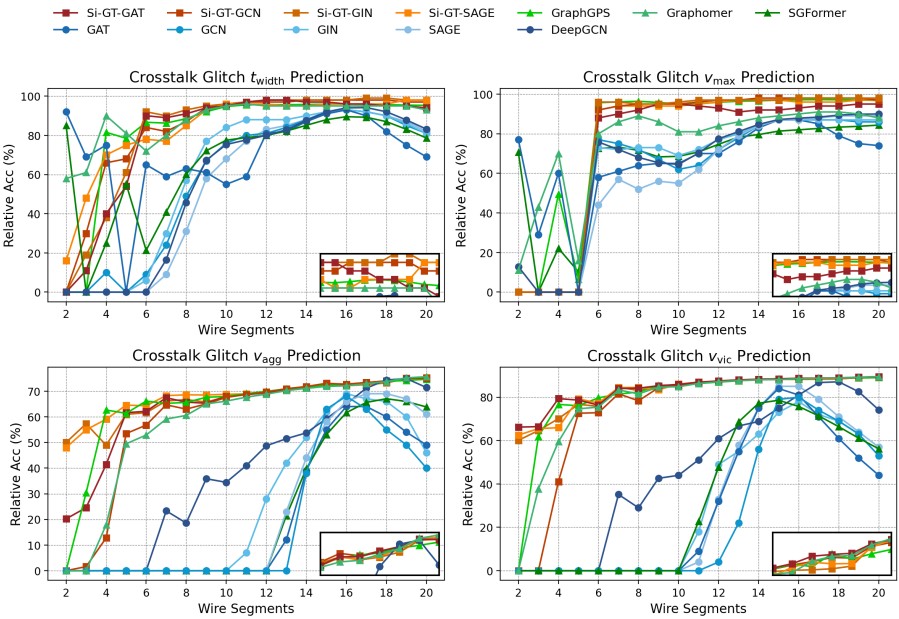

Figure 4: Comparison of models in signal integrity analysis under various IC interconnect lengths.

**Accuracy with Interconnect Length.** To illustrate the performance of our model across different interconnect scales, Figure 4 presents the model prediction accuracy with the number of wire segments of RC circuits for all prediction tasks. From the results, we can observe that: (1) Traditional GNNs exhibit notably lower accuracy for all tasks, with performance degrading on long interconnects, highlighting their inherent limitations in capturing long-range interactions critical for signal integrity analysis. (2) While DeepGCN shows improved accuracy on longer interconnects, its generalization across varying interconnect lengths remains limited. (3) Transformer-based models, particularly our Si-GT, achieve consistently higher accuracy and demonstrate robust performance even on longer interconnects. (4) All models struggle to generalize effectively to small interconnects. We analyze that short interconnects present less coupling variety, reflected in fewer data examples in dataset (detailed in Appendix A.2), resulting in poor generalization to those sparse examples.

**Segment Models in Sink-level Prediction.**
Table 4 compares the performance of segment models (trained with segment data) and sink models (trained with sink data) on sink-level prediction tasks by evaluating their differences in prediction accuracy. The results show that: (1) for victim delay prediction, training Transformer-based models with segment data can improve sink-level predictions compared to solely with sink data, while for other tasks,

Table 4: Accuracy comparison of segment and sink models in sink-level prediction tasks.

| Model | Sink Delay | | Sink Glitch | |
|---|---|---|---|---|
| | $\Delta \hat{D}_{\text{vic}}$ | $\Delta \hat{D}_{\text{agg}}$ | $\Delta t_{\text{width}}$ | $\Delta v_{\text{max}}$ |
| DeepGCN | $+6.36$ | $+11.48$ | $-2.32$ | $+0.51$ |
| SGFormer | $-2.30$ | $-12.86$ | $-3.13$ | $-6.08$ |
| Graphomer | $+0.81$ | $-1.02$ | $-1.73$ | $-5.83$ |
| GraphGPS | $+0.48$ | $+0.89$ | $-0.34$ | $-1.35$ |
| **Si-GT-GCN** | $+0.08$ | $-1.42$ | $-0.12$ | $-0.18$ |

training with sink data can yield better results; and (2) compared to baselines, the segment-trained Si-GT exhibits the smallest performance variation, demonstrating its robustness and adaptability to sink-level prediction, highlighting the robustness of Si-GT in capturing complex crosstalk behaviors even with limited structural context.

**Ablation Study.** We evaluate the impact of core design components in Si-GT through ablation studies, with results summarized in Table 5. The components under investigation include the introduction of virtual `<Net>` tokens (NET), mesh pattern encoding (MPE), and intra-inter net attention (IIN). When MPE is removed, we use centrality encoding of Graphomer to construct the input node features. In the absence of IIN, we only adopt the spatial and edge encoding of Graphomer for the attention bias in equation 1. More implementation details and fine-grained ablation experiments are provided in Appendix D.3. Our ablation study shows the critical importance of virtual `<Net>` nodes to Si-GT in all prediction tasks, as it yields a large margin performance boost in comparison

with other modules. For crosstalk delay prediction, MPE particularly shows the impact to aggressor delay prediction. Additionally, IIN attention mechanism combining both intra- and inter-net attention, consistently enhances accuracy across most tasks, which indicates that incorporating IIN encoding as an additional attention bias effectively enables the Transformer to capture the structural characteristics of coupled interconnects.

Table 5: Ablation study results on crosstalk prediction with different designs.

| Module | | | | Delay Prediction | | | | Glitch Prediction | | | |
| --- | --- | --- | --- | --- | --- | --- | --- | --- | --- | --- | --- |
| | | | | Segment | | Sink | | Segment | | Sink | |
| NET | MPE | IIN $\phi_{\text{Intra}}$ | IIN $\phi_{\text{Inter}}$ | $\hat{D}_{\text{vic}}$ | $\hat{D}_{\text{agg}}$ | $\hat{D}_{\text{vic}}$ | $\hat{D}_{\text{agg}}$ | $t_{\text{width}}$ | $v_{\text{max}}$ | $t_{\text{width}}$ | $v_{\text{max}}$ |
| × | × | × | × | 88.23 | 72.58 | 86.52 | 71.02 | 94.97 | 89.49 | 95.46 | 94.17 |
| √ | × | × | × | 88.28 | 73.30 | 87.34 | 71.04 | 98.12 | 97.70 | 97.92 | 97.57 |
| √ | √ | × | × | 88.25 | 73.48 | 87.34 | **71.93** | 98.18 | **97.85** | 98.44 | 97.90 |
| √ | √ | √ | × | 88.22 | 73.40 | 87.29 | 71.06 | 98.12 | 97.83 | 97.98 | 97.44 |
| √ | √ | × | √ | 88.27 | 73.66 | 87.26 | 70.87 | 97.97 | 97.39 | 97.99 | 97.68 |
| √ | √ | √ | √ | **88.32** | **73.67** | **87.39** | 71.82 | **98.36** | 97.78 | **98.53** | **98.63** |

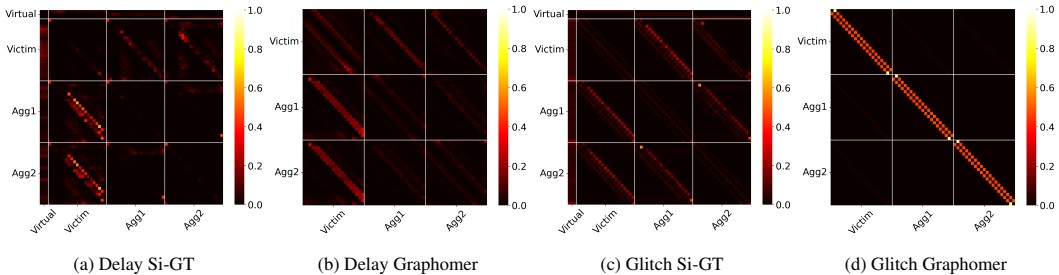

(a) Delay Si-GT  (b) Delay Graphomer  (c) Glitch Si-GT  (d) Glitch Graphomer

Figure 5: Comparison of attention maps between Si-GT and Graphomer.

**Attention Visualization.** GraphGPS applies global attention after local message passing updates, while Graphormer and our Si-GT directly integrate structural information into attention. We compare the learned attention maps of Si-GT and Graphomer in Figure 5. For delay prediction, without explicitly encoding the coupling patterns into attention bias, Graphomer (Figure 5b) shows strong attention among coupled segments, highlighting the structural importance of coupling in signal integrity analysis concerning crosstalk effect. Compared with Graphomer, Si-GT (Figure 5a) further enables the isolation of two aggressors, aligning with the fact that aggressors are not coupled with coupling capacitors in physical layout. For glitch prediction, the attention map of Si-GT (Figure 5c) shows clear coupling pattern, while Graphomer (Figure 5d) only concentrates on the neighbor nodes of the same net, limiting its ability to model noise propagation across coupled nets.

**Computation Efficiency.** We compare the computational efficiency of promising Transformer-based model Graphomer, GraphGPS, and our Si-GT model against SPICE simulation in this section. All reported runtimes are measured on CPU. Details of the computing environment and additional runtime benchmarks across different hardware platforms are provided in Appendix C.2. As shown in Figure 6, the computational cost of SPICE increases substantially with interconnect length, while Transformer-based models maintain consistently low inference times. On average, Graphomer, GraphGPS, and Si-GT achieve inference times of 2.4 ms, 6.8 ms, and 4.0 ms, respectively, compared to over 100 ms required by SPICE even for short interconnects, highlighting the practicality of transformer-based models as efficient and scalable alternatives for signal integrity analysis in large-scale IC designs.

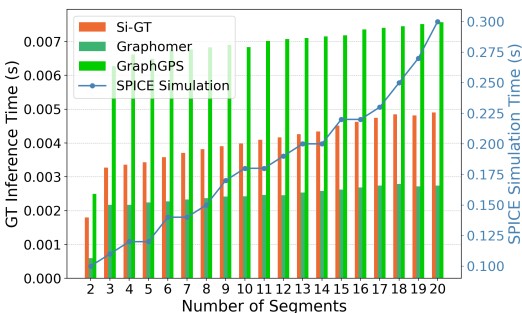

Figure 6: SPICE vs. Transformer-based models running time across varying interconnect scales.

## 6 CONCLUSION

In this paper, we propose Si-GT, a Transformer-based model for signal integrity analysis of IC interconnects, and construct the first large-scale benchmark dataset comprising crosstalk prediction tasks relevant to practical SI challenges. We demonstrate that Si-GT consistently outperforms state-of-the-art GNN and GT baselines across nearly all tasks, while significantly reducing runtime compared to SPICE. These results highlight the strong potential of Si-GT as an efficient surrogate for interconnect signal integrity analysis to accelerate IC design verification.

## 7 ACKNOWLEDGEMENT

This work was supported, in part, by the SUNY-IBM AI Collaborative Research Alliance, the SUNY Empire Innovation Program, and the Graduate School "Intelligent Methods for Test and Reliability" at the University of Stuttgart.

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

# A  DATASET DETAILS

## A.1  DATASET PREPARATION

Given the separation between adjacent wires, as the length, width, and the relative electric permittivity $\varepsilon r$ of the substrate are constant per wire segment, we calculate the coupling capacitance according to Eq.6, based on the two-dimensional capacitance formula from (Sakurai & Tamaru, 1983).

$$C_c = \varepsilon r * \frac{\text{Segment Length of Wire * Wire Width}}{\text{Wire Seperation}} \tag{6}$$

With the sweeping parameter setting in Table 1, we use Algorithm 1 to create the netlist of various RC circuits modeling coupled interconnects. After the netlist is created, we supply a pulse input with 0.7 magnitude to every active interconnect wire. SPICE simulation is carried out to measure the voltage waveforms at each segment along individual nets. Only examples that successfully complete SPICE simulations without failures are retained in the dataset.

---

**Algorithm 1** Generate RC netlist for coupled interconnects

---

**Require:** $wr$: wire resistance per micron, $wc$: wire capacitance per micron, $\{C_c\}$: set of coupling capacitance values; $\{l\}$: wire segment length.
    Sample number of segments $N \sim \mathcal{U}(2, 20)$
    Initialize segment index $s \leftarrow 1$
    Set wire segment length: $l \leftarrow 5\mu m$
    **while** $s \leq N$ **do**
        Set wire resistance: $R_w \leftarrow l \cdot wr$
        Set wire capacitance: $C_w \leftarrow l \cdot wc$
        Sample coupling capacitance between victim and aggressor 1: $\hat{C}_s \sim \text{Random Select}(\{C_c\})$
        Sample coupling capacitance between victim and aggressor 2: $\hat{C}_s \sim \text{Random Select}(\{C_c\})$
        $s \leftarrow s + 1$
    **end while**
    =0

---

## A.2  DATASET DISTRIBUTION

To show the composition of our signal integrity dataset, we analyze the distribution of examples across different wire segments. Figure 7 illustrates the number of examples for both the crosstalk delay (Figure 7a) and crosstalk glitch (Figure 7b) prediction tasks.

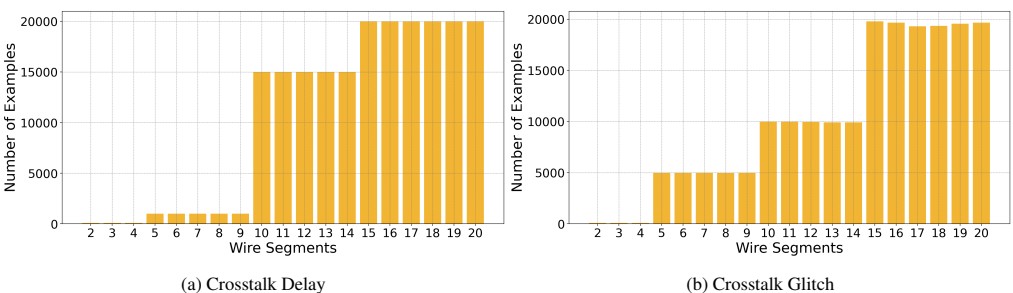

(a) Crosstalk Delay          (b) Crosstalk Glitch

Figure 7: Distribution of the signal integrity dataset across wire segments.

# B  GRAPH FEATURIZATION.

**Directed or Undirected.** As illustrated in Figure 2, we construct the interconnect graph by modeling edges within each individual net as directed from source to sink. In this section, we evaluate

model performance using undirected interconnect graphs as input, aiming to assess the impact of edge directionality on prediction accuracy. We primarily report the prediction accuracy difference between undirected and directed settings on segment-level prediction tasks for both aggressor and victim nets for comparison. The results are reported in Table 6. A positive difference indicates improved performance with undirected graphs, while a negative difference suggests that maintaining directionality is beneficial for capturing the underlying signal behavior in the circuit.

Table 6: Comparison of segment-level prediction accuracy using directed vs. undirected interconnect graphs.

| Model | Undirected | | | |
|---|---|---|---|---|
| | Segment Delay | | Segment Glitch | |
| | $\Delta \hat{D}_{\mathrm{vic}}$ | $\Delta \hat{D}_{\mathrm{agg}}$ | $\Delta t_{\mathrm{width}}$ | $\Delta v_{\mathrm{max}}$ |
| GCN | +3.80 | +3.62 | +1.05 | −0.10 |
| DeepGCN | +7.98 | +11.48 | −0.10 | −1.08 |
| Graphomer | +1.89 | +1.26 | −0.82 | −3.28 |
| GraphGPS | −1.35 | −0.28 | +0.39 | −1.53 |
| **Si-GT-GCN** | −0.13 | −0.20 | +2.30 | −0.16 |

## C  EXPERIMENT DETAILS

### C.1  DETAILED EXPERIMENTAL SETTINGS

For the model training, we use different training schemes for the models included in our experiments:

**Standard GNN models.** We train the model using the Adam optimizer with a learning rate of 2e-3 and a weight decay of 6e-4. Models are trained with 256 batch size for 100 epochs.

**DeepGCN.** We train the model using the Adam optimizer with a learning rate of 1e-3 and a weight decay of 6e-4. Models are trained with 256 batch size for 100 epochs.

**GraphGPS.** We train the model using the Adam optimizer with a learning rate of 5e-4 and a weight decay of 1e-5. Models are trained with 256 batch size for 100 epochs.

**SGFormer.** We train the model using the Adam optimizer with a learning rate of 5e-5 and a weight decay of 1e-5. Models are trained with 256 batch size for 200 epochs.

**Graphomer and Si-GT.** We train the model using the AdamW optimizer with an initial learning rate of 1e-4, polynomial learning rate decay, and linear warmup, where the learning rate decays to 1e-9 over the total training steps, with weight decay set to 1e-4. Models are trained with 256 batch size for 60 epochs.

All experiments in this paper are implemented with PyTorch 2.2.2, DGL 2.4.0, and Pytorch-geometric 2.7.0.

### C.2  COMPUTING ENVIRONMENT

All models are trained with $2\times$ NVIDIA A100 80GB GPUs. SPICE simulations are carried out with the commercial Synopsys HSPICE simulator on an Intel Core i7-11700K Processor. In Figure 6, we report the running time of Transformer-based models executed on an Intel Xeon Gold 6448Y Processor. Additionally, we compare the running time of Graphomer, GraphGPS, and Si-GT on the A100 GPU in Figure 8. As our graph sizes are relatively small, GPU inference may exhibit higher latency due to kernel launch overhead and underutilization of GPU parallelism.

### C.3  BASELINE IMPLEMENTATION.

For crosstalk delay prediction, the model outputs a single delay value, so the output dimension is set to 1. For crosstalk glitch prediction, we predict both the peak voltage and noise width, so the output dimension is set to 2. The model architectures of the baselines are summarized as follows:

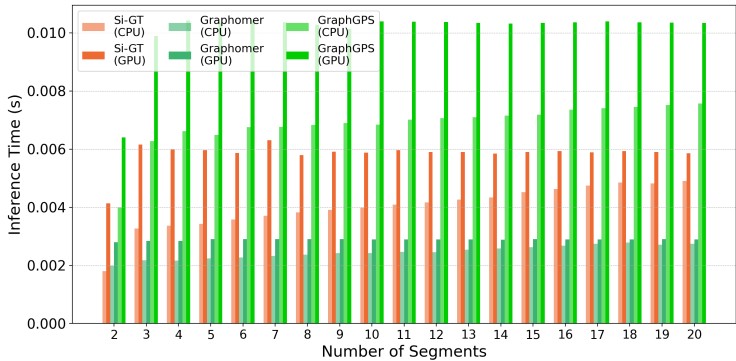

Figure 8: Model inference time.

**Standard GNN models.** We set convolutional layers to 5 and the hidden dimension to 64 for glitch prediction and 128 for delay prediction. Each convolutional layer is followed by ReLU activation and PairNorm normalization, except for the final layer.

**DeepGCN.** We set convolutional layers to 20 and the hidden dimension to 64 for glitch prediction and 128 for delay prediction. It stacks 20 DeepGCNLayer blocks, each composed of a GCNConv layer for message passing, LayerNorm for normalization, a ReLU activation, and 0.1 dropout. These blocks use a configurable residual connection strategy to enable stable training of deep GNNs (Li et al., 2019).

**SGFormer.** SGFormer integrates a GCN-based GraphModule and a Transformer-style SGModule. In our implementation, we set the hidden dimension to 64. The SGModule uses 2 transformer layers, each with 4 attention heads and 0.5 dropout, to model long-range interactions via dense attention. In parallel, the GraphModule applies 3 layers of GCNConv with dropout 0.5 and residual connections to extract localized features. The outputs from both modules are averaged to form the final node representation.

**Graphomer.** We use 6 encoder layers with a hidden dimension of 64 and 4 attention heads; each layer includes a feed-forward network with an embedding dimension of 128, and a dropout rate of 0.1 is applied after multi-head self-attention. Graphomer uses the shortest path between any pair of nodes for spatial and edge encoding. Follow (Ying et al., 2021), the length limit of the shortest path is set to 5 by default.

**GraphGPS.** For the PE/SE of GraphGPS, we use RSWE with a walk length of 16 by default. In our implementation, the input node features are projected to 64 dimensions, with 16 dimensions for PE/SE. The model stacks 10 GPSConv layers, each integrating a GINEConv-based local aggregator and multi-head attention mechanism with 4 heads to capture global interactions. A 3-layer MLP with decreasing dimensions is applied for the final prediction.

**Si-GT.** We use 6 encoder layers with a hidden dimension of 64 and 4 attention heads; each layer includes a feed-forward network with an embedding dimension of 128, and a dropout rate of 0.1 is applied after multi-head self-attention. For mesh pattern encoding, we use 2 convolutional layers (e.g., EGATConv, GraphConv, SAGEConv, GINConv in DGL.) with residual connection, and we set 0.2 dropout rate for node embeddings.

We report the trainable model parameters of all models in Table 7.

Table 7: Trainable parameter size of models.

| Model | GCN | GAT | GIN | SAGE | DeepGCN | SGFormer | Graphomer | GraphGPS | Si-GT GCN | Si-GT GAT | Si-GT GIN | Si-GT SAGE |
|---|---|---|---|---|---|---|---|---|---|---|---|---|
| Parameter Size | 49,921 | 52,486 | 115,971 | 99,329 | 85,953 | 210,561 | 273,261 | 422,417 | 282,029 | 306,733 | 282,029 | 290,221 |

## C.4 TRAINING CURVE

We plot the training loss curves for crosstalk delay and glitch prediction tasks in Figure 9. Across both tasks, Si-GT consistently achieves faster convergence and lower final training loss compared to other models.

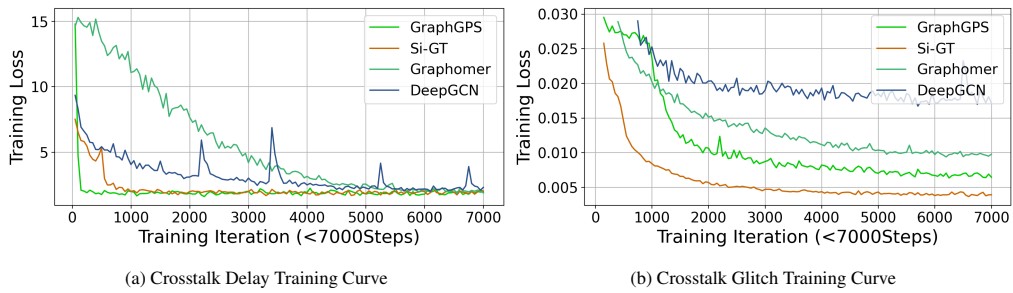

(a) Crosstalk Delay Training Curve       (b) Crosstalk Glitch Training Curve

Figure 9: Training curve of models for signal integrity analysis tasks.

# D MORE EXPERIMENTS

## D.1 MORE EXPERIMENTS WITH DIFFERENT STRUCTURAL ENCODING

In this section, we conduct experiments with random walk structural encoding (RWSE) and Laplacian-based positional encoding (LapPE) for GraphGPS and our Si-GT. Specifically, we replace the mesh pattern encoding in Si-GT with RWSE and LapPE variants, and compare the performance against GraphGPS. In the experiments, we vary the random walk length with 4, 8, 16 for RWSE-based position encoding of GraphGPS model (e.g., RWSE16) and set 8 top eigenvectors of the graph Laplacian for LapPE (e.g., LapPE8). Additionally, we compare Si-GT with GraphGPS using composite position encoding (e.g., LapPE8+RWSE16), following the implementation in (Rampášek et al., 2022), we concatenate the LapPE and RWSE vectors to form the final positional encoding. Experimental results are reported in Table 8.

Table 8: Mean relative accuracy (%) of crosstalk delay and glitch prediction results.

| Model | PE/SE | Segment Delay $\hat{D}_{vic}$ | $\hat{D}_{agg}$ | Segment Glitch $t_{width}$ | $v_{max}$ |
|---|---|---|---|---|---|
| GraphGPS | LapPE8 | 87.76 | 71.69 | 96.15 | 97.30 |
| GraphGPS | RWSE4 | 88.12 | 72.19 | 95.67 | 97.21 |
| GraphGPS | RWSE8 | 88.24 | 71.97 | 95.14 | 97.69 |
| GraphGPS | RWSE16 | 88.23 | 72.65 | 96.61 | **97.99** |
| GraphGPS | LapPE8+RWSE16 | 88.28 | 72.92 | 96.49 | 97.83 |
| **Si-GT** | LapPE8 | 87.72 | 73.50 | 95.25 | 93.18 |
| **Si-GT** | RWSE16 | 88.21 | **73.78** | 97.44 | 96.30 |
| **Si-GT** | LapPE8+RWSE16 | 87.85 | 73.47 | 96.87 | 95.32 |
| **Si-GT** | MPE | **88.32** | 73.67 | **98.36** | 97.78 |

As shown in Table 8, Si-GT consistently achieves higher accuracy in delay and glitch width prediction tasks compared to GraphGPS across various positional encoding configurations. These results highlight the effectiveness of our mesh pattern encoding (MPE) and demonstrate the robustness of Si-GT when combined with both LapPE and RWSE encodings.

## D.2 INFERENCE EXAMPLES

We visualize the predicted values of key signal integrity metrics against the SPICE-measured ground truth in Figure 10. For crosstalk glitch and crosstalk delay of victim prediction tasks, Si-GT consistently provides the closest match to the ground truth across all metrics, demonstrating its ability

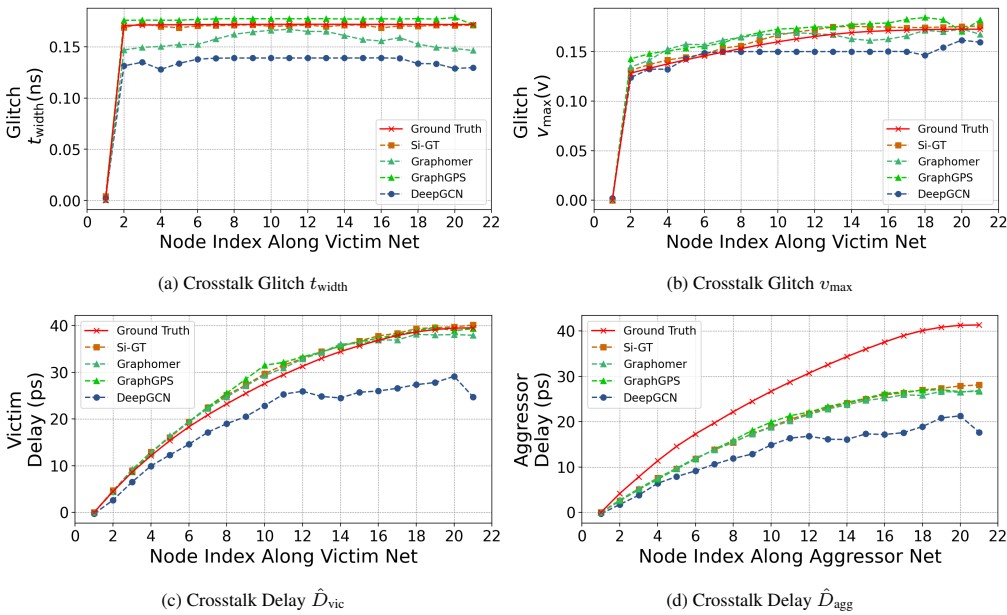

(a) Crosstalk Glitch $t_{\text{width}}$

(b) Crosstalk Glitch $v_{\text{max}}$

(c) Crosstalk Delay $\hat{D}_{\text{vic}}$

(d) Crosstalk Delay $\hat{D}_{\text{agg}}$

Figure 10: Comparison of crosstalk prediction accuracy with the number of wire segments.

to accurately model both local coupling effects and global signal dependencies. Additionally, we present an example of aggressor delay prediction. Since capacitive coupling primarily affects the victim net, signal integrity analysis mainly focuses on victim-side behavior.

## D.3 FINE-GRAINED ABLATION EXPERIMENTS

**Ablation experiment setup.** Table 5 summarizes the ablation results for the core architectural components introduced in Si-GT. All ablations are performed using the strongest Si-GT backbone selected for each prediction task, e.g., GCN for delay-segment, GAT for delay-sink, GIN for glitch-segment, and GCN for glitch-sink, following the configurations reported in Table 2 and Table 3.

**Additional analyses.** To further understand the contribution of each component, we conduct more fine-grained ablation studies. In Equation 4, spatial encoding and edge encoding are incorporated as attention bias terms to better capture the global structural context of interconnect topology. In this section, we remove $\tilde{\Phi}$d and $\tilde{\Phi}$sp from the attention bias to isolate their effect on model performance.

Table 9: Ablation study on the removal of spatial and edge encoding biases.

| Model | Delay Prediction | | | | Glitch Prediction | | | |
| | Segment | | Sink | | Segment | | Sink | |
| | $\hat{D}_{\text{vic}}$ | $\hat{D}_{\text{agg}}$ | $\hat{D}_{\text{vic}}$ | $\hat{D}_{\text{agg}}$ | $t_{\text{width}}$ | $v_{\text{max}}$ | $t_{\text{width}}$ | $v_{\text{max}}$ |
|---|---|---|---|---|---|---|---|---|
| Si-GT-without $\tilde{\Phi}$d, $\tilde{\Phi}$sp | 87.25 | 72.84 | 87.03 | 71.50 | 97.74 | 97.13 | 97.59 | 97.57 |
| Si-GT | 88.32 | 73.67 | 87.39 | 71.82 | 98.36 | 97.78 | 98.53 | 98.63 |

