# OpenReview forum: "Si-GT: Fast Interconnect Signal Integrity Analysis for Integrated Circuit Design via Graph Transformers"
_ICLR.cc/2026/Conference — ICLR 2026 Poster_

### Official Review · Reviewer_DwGg · 2025-10-29

**Soundness:** 2
**Presentation:** 2
**Contribution:** 2
**Rating:** 6
**Confidence:** 3

**Summary:**

The paper is trying to make better Graph transformers for modeling analog-level effects in circuits, which is quite a pressing problem. ML approaches for digital IC are developing, but for analog/mixed-signal-level interference and crosstalk, the efforts are in a nascent stage. The approach seems to outperform other graph-network like approaches for the problem, and they also make their own simulated dataset based on ground truth labels via synopsys tools.

**Strengths:**

Analog / mixed-signal modeling via ML models is understudied, and datasets are also hard to make and standardize. The paper takes steps towards both, and makes some effort to customize vanilla GT to make Si-GT work. Ablations seem reasonable as well.

**Weaknesses:**

There are some major weaknesses with this paper, stemming from setup, empirical eval, and industrial relevance.

Setup : It is not clear at all why graph transformers or for that matter graph networks are the right lens to study this kind of circuit. Yes, it can be represented as a graph, but it is a highly regular, patterned graph. One can represent an image as a grid graph, but that does not make the graph neural network ideal for images. Why is it obvious that we need to solve this problem via GT ?

Eval : The dataset is mostly simulation-level, and it's very unclear that it reflects the industrial circuits (I understand that the standards are Intel/Synopsys based, but that does not mean the random sweep generates circuits relevant to industry). Further, the paper is evaluating its own methods on its own dataset, which makes it impossible to gauge impartially. (I do agree it is kind of a chicken and egg problem, which is why selecting pre-existing datasets will ease the question of impartiality). The baselines are fairly ad hoc and "generic" GT/GNN models, which may not be suitable. The layers also look a bit deep for this problem (GNNs are well known to suffer with depth)

Relevance : In analog or digital IC, there are already pre-existing tools that can create the labels in question (indeed the paper uses them). The abstract says that this is not always feasible due to computational costs (true) but then the question becomes : what is the computational tradeoff of the papers' models vs running the tool itself ? This is not studied. We are of course ignoring the fact that even slight losses in accuracy relative to industrial standards are generally considered unacceptable.

**Questions:**

Please see the "weaknesses" section - in particular - the questions about industrial relevance and benchmarking to other methods.

---

> ### Author Response · Authors · 2025-11-22
>
> We greatly appreciate the reviewer for the comments. We respond to each comment below and sincerely hope that our responses could properly resolve your concerns.
>
> ## Review1: The choice of graph transformers as the modeling framework.
> Crosstalk delay and glitch arise from distributed RC interactions and signal propagation along the wire path, where both within-net dependencies (accumulated resistance and downstream loading) and cross-net couplings (capacitive aggressor–victim interactions) jointly determine the final waveform. These effects induce long-range, non-local dependencies that cannot be captured by local operators such as convolutional filters, unlike image grids where spatial locality dominates. In contrast, the interconnect structure is naturally represented as a graph: each net forms a propagation path, and coupling capacitances introduce explicit cross-net edges. The resulting dependency patterns are directional rather than a regular grid, making graph-based modeling a more faithful abstraction of the underlying physics.
>
> Our empirical results in Tables 2–3 and Figure 4 further validate this perspective. Traditional GNNs underperform, especially on long nets, due to over-squashing and their limited ability to propagate information over extended signal paths. Graph Transformers address these limitations by enabling global attention over the entire propagation tree and by incorporating physically meaningful attention biases (e.g., resistance accumulation and coupling strength). These capabilities are essential for capturing long-range propagation and pattern-dependent crosstalk behavior, which are central to accurate delay and glitch prediction. Therefore, employing graph transformers is not arbitrary, but follows directly from the physical structure of the problem and is empirically supported by our comparisons.
>
> ## Review2: Industrial relevance of dataset.
> Direct access to industrial post-routing netlists with full parasitics is typically restricted due to IP and confidentiality constraints. We agree that using proprietary design netlists would further strengthen practical relevance. However, the underlying crosstalk mechanisms are governed by modeled RC network interactions that are typically solved with SPICE simulations in industrial standard. To ensure realism, we generate all examples using SPICE with parasitic parameters aligned to Intel 14 nm technology and adhering to industry-standard RC models for interconnect segments.
>
> We introduce variation by systematically sweeping wire spacing, segment length, coupling capacitance, slew rate, and switching direction parameters that are routinely varied in SI corner analysis. This procedure enables comprehensive coverage of aggressor–victim scenarios and reflects the range of behaviors encountered in real post-routing crosstalk conditions.
>
> ## Review3: Baseline choice.
> Regarding baselines, we deliberately selected strong, representative graph models, including advanced GTs ( Graphomer, SGFormer, GraphGPS), because they are state-of-the-art hybrid and transformer-based graph models designed for long-range dependency modeling. They are the closest architectural analogues. Our experiments (Tables 2–3) show that GraphGPS is indeed competitive and achieves performance close to Si-GT on certain tasks.
>
> On the depth question, we emphasize that traditional GNNs do suffer when naively stacked due to over-squashing and vanishing expressivity. This is precisely why we use a graph-transformer backbone: global attention mitigates the depth-related issues inherent to message-passing GNNs and is crucial for modeling long-range RC interactions across extended interconnect paths.
>
> ## Review4: Computational tradeoff of Si-GT vs SPICE.
>
> Si-GT offers a substantial runtime improvement while maintaining high accuracy. Figure 6 provides a direct comparison between SPICE and Transformer-based models. SPICE runtime grows linearly with wire length and can exceed 100 ms even for short nets. Si-GT inference averages ~4 ms, representing >20× speedup. Graphomer and GraphGPS fall in a similar range, confirming that transformer-based surrogates offer consistent speed advantages over circuit simulation.
>
> Industrial sign-off tools require extremely accurate SI estimation, which is why our prediction tasks target SPICE-level measurements (delay and glitch at segment- and sink-level). Finally, surrogate models are not intended to replace sign-off simulators, but to accelerate early-stage SI analysis, reduce the number of SPICE runs needed for convergence, and provide fast sensitivity feedback to designers. This aligns with current industry practice for surrogate modeling (e.g., ML-based timing and parasitics prediction).

---

> > ### Comment · Reviewer_DwGg · 2025-11-26
> > **Acknowledgement of rebuttal**
> >
> > I have read the authors' rebuttal and it broadly addresses my concerns. I will maintain my score as leaning towards acceptance.

---

### Official Review · Reviewer_kjJC · 2025-10-30

**Soundness:** 2
**Presentation:** 3
**Contribution:** 2
**Rating:** 2
**Confidence:** 5

**Summary:**

This work proposed a fast interconnect signal integrity analysis by graph transformers. Their work modeled crosstalk effects explicitly with aggressor-victim switching interactions and signal pattern-dependent analysis, which are missing in the current works. They verified their ideas on the artificial dataset, whose samples were generated based on two aggressors and one victim.

**Strengths:**

1. Good modeling of the signal integrity analysis problem. Their work modeled crosstalk effects explicitly with aggressor-victim switching interactions and signal pattern-dependent analysis, which are missing in the current works. The consideration about net-specific signal characteristics (e.g., switching direction and slew
rate) is meaningful for the prediction of crosstalk delay and glitch.
2. The experiment shows more efficiency when compared with the simulation tool, e.g., SPICE.

**Weaknesses:**

1. The artificial dataset deviates from the realistic circuit design. Although the authors generate a massive dataset, they rely on only three nets: two aggressors and one victim. In the integrated circuit design, even the smallest circuits have more than three nets to build a specific function. I am concerned about the usability of this method in a realistic design flow.
2. The experiments mainly compare with some classic graph learning algorithms, lacking comparison with other ML-based signal integrity analysis methods. In Section 2, the authors have a survey of related works in "ML for SI" and criticize them for not explicitly modeling signal pattern variability. But I still think this paper should be compared with the related works in some metrics, e.g., sink delay.

**Questions:**

1. Why don't the authors compare their work with the related works in Section 2? I think the comparison will enhance the credibility of this work.
2. The method in this work, whose goal is to reduce the computation of SPICE, may be more suitable for signal integrity analysis in the standard cell design. I suggest the authors perform experiments based on a standard cell library, e.g., ASAP7 or other commercial libraries. I wonder if the authors have plans to test their methods on this type of dataset?

---

> ### Author Response · Authors · 2025-11-21
>
> We greatly appreciate the reviewer for the comments. We respond to each comment below and sincerely hope that our responses could properly resolve your concerns.
> ## Review1: Dataset limitation.
> We thank the reviewer for raising this important point and address the concern from the following aspects:
> ### (1) Dataset realism.
> * **Crosstalk is fundamentally local.**
> Crosstalk effects are dominated by immediately adjacent neighbors. Although a net may connect to many nets globally, the effective coupling neighborhood of any wire segment typically includes only its 1–2 closest parallel neighbors due to the rapid spatial decay of capacitive coupling.
>
> * **Industry-standard practice.**
> Commercial sign-off tools compute coupling-induced delta-delay using pairwise aggressor–victim interactions. In real flows, SPICE configurations and timing-window filtering further reduce the number of effective aggressors.
>
> * **Analytical and ML-based models.**
> Classical analytical SI models [1-5]  and recent ML-based SI methods such as GraphCAD [6]  also employ the same pairwise aggressor–victim abstraction because it reflects the underlying coupling physics.
>
> ### (2) Dataset comparison with GraphCAD [6].
> GraphCAD uses real post-route designs and reports overall design statistics in Table 3 of their paper. However, the number of pronounced crosstalk cases is not revealed. From the test comparisons in Table 4, the number of RC-VA graphs corresponding to victim–aggressor cases remains small in scale. Regarding the task objective:
> * **GraphCAD** predicts sink-level SI-aware delay, and does not incorporate diverse driving-pattern combinations in either its model or dataset design.
> * **Our dataset** targets both crosstalk delay and crosstalk glitch prediction at both segment level and sink level under multiple signal-pattern combinations. Our model Si-GT is explicitly designed to capture signal patterns by the virtual NET token.
>
> Although our dataset uses a three-net configuration, it remains physically realistic because we perform sweeps across parameters that dominantly influence crosstalk, ensuring coverage across key physical and leads to a dataset that is both larger in scale and richer in SI variability than GraphCAD.
>
> ## Review2: Compare with related works.
> We appreciate the reviewer’s suggestion. The most relevant existing work to ours is GraphCAD [6]. However, its dataset and model implementations are not open-source, making it impossible to test our model fairly against their setup. Nevertheless, we clarify the conceptual differences that make a direct comparison nontrivial:
>
> * Regarding the task scope, GraphCAD focuses exclusively on predicting sink-level crosstalk-affected delay. It does not consider diverse signal patterns and glitch scenarios, which are critical for realistic signal integrity analysis.
>
> * Regarding the model design, while this method provides a structured way to model heterogeneous interactions, it introduces a strong separation between coupling segments. Specifically, in the Graph Transformer input stage of their model, aggressor and victim wire paths are considered unconnected; they introduce an extra graph attention network to combine the representations of individual nets, rather than through their actual physical topologies. This design choice overlooks the fact that local coupling can induce long-range effects as waveforms propagate along the interconnect, limiting its ability to capture pattern-dependent glitch shaping and distributed crosstalk behavior.
>
> We would be glad to include a sink-delay comparison once the GraphCAD model and dataset become publicly available. At present, we ensure fairness by comparing against reproducible graph-learning baselines and will clarify this point more explicitly in the paper revision.
>
> ## Review3: Apply Si-GT for SI in standard cell design.
> We thank the reviewer for this valuable suggestion. Our goal is to provide a fast and accurate surrogate for SPICE-based SI analysis, and our current dataset is designed to capture the underlying physics of crosstalk behavior through controlled parametric sweeps. We agree that applying our method to standard cell–based designs is important, and we welcome any opportunities to build datasets with commercial libraries and evaluate our method as part of future extensions.
>
> **References**
>
> [1] A. Vittal et al., “Crosstalk in VLSI interconnections,” IEEE TCAD, 18(12), 1999.
>
> [2] S.-C. Wong, G.-Y. Lee, D.-J. Ma, “Modeling of interconnect capacitance, delay, and crosstalk,” IEEE TSM, 13(1), 2000.
>
> [3] W.-Y. Chen, S. Gupta, M. Breuer, “Analytical models for crosstalk excitation and propagation,” IEEE TCAD, 21(10), 2002.
>
> [4] M. Kuhlmann, S. S. Sapatnekar, “Exact and efficient crosstalk estimation,” IEEE TCAD, 20(7), 2001.
>
> [5] K. Takeuchi et al., “Probabilistic crosstalk delay estimation for ASICs,” IEEE TCAD, 23(9), 2004.
>
> [6] F. Liu et al., “GraphCAD: Graph Neural Networks for Crosstalk-affected Delays,” ISPD, 2025.

---

> > ### Comment · Reviewer_kjJC · 2025-11-28
> > **Reply to comments.**
> >
> > Thanks to the reviewer for addressing my concerns. I realize the dataset setup is consistent with commercial tools, since crosstalk analysis in those tools is also a localized issue. Meanwhile, the method in this paper is more conceptually sound than its ML-based counterpart, GraphCAD, though a fair experimental comparison is somewhat difficult. Therefore, I will raise my ratings for this paper. But it seems the rating button is not working now.

---

### Official Review · Reviewer_Eseo · 2025-10-30

**Soundness:** 3
**Presentation:** 3
**Contribution:** 2
**Rating:** 4
**Confidence:** 3

**Summary:**

This paper proposes Si-GT, a novel Graph Transformer model for fast and accurate signal integrity (SI) analysis in integrated circuit (IC) interconnects, specifically targeting crosstalk-induced delay and glitch prediction. The main contributions are:
1. Model Design: Si-GT introduces three key components to embed physical priors into the Transformer architecture: (a) a virtual <NET> token to encode net-level signal characteristics (e.g., switching direction, slew rate); (b) Mesh Pattern Encoding (MPE) to capture local coupling structures at each node; and (c) an Intra-Inter Net (IIN) attention mechanism to explicitly model both intra-net signal propagation and inter-net capacitive coupling.
2. Dataset Construction: The authors construct a large-scale dataset with 200,200 crosstalk delay examples and 187,309 crosstalk glitch examples, using SPICE simulations as ground truth. They claim this is the first dataset dedicated to interconnect SI analysis.
3. Empirical Validation: Experiments show that Si-GT outperforms state-of-the-art GNN and Graph Transformer baselines in prediction accuracy while achieving orders-of-magnitude speedup over SPICE.

**Strengths:**

1. Problem Significance: The paper addresses a critical and practical challenge in modern IC design—crosstalk-induced signal integrity degradation, which directly impacts chip reliability and timing closure.
2. Physics-Informed Architecture: The proposed designs (<NET> token, MPE, IIN) are well-motivated by circuit physics and aim to inject domain-specific inductive bias into the model, moving beyond generic graph learning.
3. Potential Community Impact: If made publicly available, the dataset could serve as a valuable benchmark for future research in ML for EDA, especially for crosstalk-aware modeling.

**Weaknesses:**

1. Dataset Limitations and Lack of Clarity:
(1)	The paper does not state whether the dataset will be open-sourced, which is essential for reproducibility and community adoption.
(2)	The synthetic data is based on a highly simplified topology of “2 aggressors + 1 victim”, which fails to reflect the complex multi-net coupling scenarios in real VLSI layouts. In contrast, recent work like GraphCAD (ISPD '25) uses real post-routing circuits from design contests, offering far greater realism and diversity.
(3)	There is no quantitative comparison with statistics from prior datasets (e.g., number of nets, coupling density, net length distribution), weakening the claim of being “large-scale” and representative.
2. Inadequate Baseline Comparisons:
(1)	The paper omits comparison with task-specific SOTA models, notably GraphCAD (2025) and Routing-Free Crosstalk Prediction (Liang et al., 2020) , both of which directly address crosstalk modeling. This is a major oversight.
(2)	Si-GT uses a “GNN → Transformer” pipeline, yet it does not compare against generic GNN-Transformer hybrids (e.g., GraphTrans or GraphGPS variants with GNN preprocessing), making it unclear whether gains stem from architecture or the proposed inductive biases.
3. Questionable Experimental Design:
(1)	In ablation studies (Table 5), different Si-GT variants (e.g., Si-GT-GCN, Si-GT-GAT) use different GNN backbones, introducing confounding variables. A fair ablation should fix the GNN type.
(2)	The performance gains over strong baselines like GraphGPS are marginal, raising questions about the practical significance of the added complexity.
(3)	The paper adds Graphormer-style shortest-path (SP) and edge biases on top of IIN, but provides no justification for this design choice or analysis of potential redundancy/conflict between SP bias and the intra-net resistance-based bias.
4. Presentation Issues:
(1)	Figure 3 (model overview) is abstract and poorly aligned with the text, reducing clarity.
(2)	Experimental details referenced in the main text (e.g., ablation setup) are missing in the appendix, undermining reproducibility.

**Questions:**

1. Dataset Openness and Comparison: Will the proposed dataset be publicly released? If so, please confirm. Additionally, please provide a detailed statistical comparison (e.g., net count, coupling complexity, segment distribution) with datasets used in prior works such as Liang et al. (2020) and Liu et al. (2025).
2. Missing Baselines: Why were GraphCAD (Liu et al., 2025) and Routing-Free Crosstalk Prediction (Liang et al., 2020) not included as baselines? Please add experimental results comparing Si-GT against these task-specific SOTA models.
3. Attention Bias Design: The model combines IIN bias with Graphormer’s shortest-path (SP) and edge biases. What is the motivation for this combination? Specifically, could the SP bias conflict with \phi_{Intra}, which already encodes distance via accumulated resistance ? Please clarify through ablation or analysis.
4. Granular Ablation of IIN: The current ablation removes the entire IIN module. Please perform a finer-grained ablation by separately disabling \phi_{Intra} and \phi_{Inter} to quantify their individual contributions.
5. Architecture Fairness: Since Si-GT uses GNN features as input to the Transformer, please compare against GraphGPS or GraphTrans variants that also use GNN-preprocessed features, to isolate the benefit of your proposed components (MPE, <NET>, IIN) from the general “GNN+Transformer” paradigm.
[1] Liu et al., GraphCAD: Leveraging graph neural networks for accuracy prediction handling crosstalk-affected delays, ISPD ’25.
[12] Liang et al., Routing-Free Crosstalk Prediction, ICCAD 2020.

---

> ### Author Response · Authors · 2025-11-21
>
> We greatly appreciate the reviewer for the comments. We respond to each comment below and sincerely hope that our responses will properly resolve your concerns.
> ## Review1: Comparison with Routing-free Crosstalk and GraphCAD.
> - **Routing-free crosstalk:** Identifies crosstalk-critical nets during placement without routing information, addressing a different task.
>
> - **GraphCAD:** Predicts sink-level crosstalk delay after routing, but it uses only one worst-case switching pattern (maximum victim rise, maximum aggressor fall), without modeling pattern variability or glitch behavior.
>
> - **Si-GT:** Predicts both crosstalk delay and glitch at segment and sink levels after routing under diverse switching patterns (direction, slew, active/quiet states).
>
> ## Review2: Dataset statistical comparison.
> We will release the dataset and code upon acceptance. Although GraphCAD is relevant, it only reports the design statistics, and the actual number of victim–aggressor cases extracted from designs is not provided. From the test comparisons in Table 4 of the GraphCAD paper, the number of RC-VA graphs corresponding to victim–aggressor cases remains small in scale.  Because the task scopes differ, a direct statistical comparison would not be meaningful.
>
> ## Review3: Missing baseline.
> Routing-Free Crosstalk and GraphCAD are not publicly available, and their task formulations differ substantially from Si-GT. Reproducing them without access to their datasets or parameters would risk unfair or misleading comparisons.
>
> ## Review4: Ablation study.
> ### (1) A fair ablation should fix the GNN type.
> In Table 5, the GNN backbone is fixed within each prediction task. The backbone only differs across tasks because these tasks empirically benefit from different local encoders. Our ablations are conducted using the strongest Si-GT configuration for each task e.g., GCN for delay-segment, GAT for delay-sink, GIN for glitch-segment, and GCN for glitch-sink, as shown in Table 5.
>
> ### (2) Granular ablation of IIN attention.
> We perform ablations of the IIN module by disabling IntraNetAttn and InterNetAttn independently. The results are as follows:
> | Ablation Configuration | Segment-$\hat{D}_\text{vic}$ | Segment-$\hat{D}_\text{agg}$ | Sink-$\hat{D}_\text{vic}$ | Sink-$\hat{D}_\text{agg}$ |
> |:---|:---:|:---:|:---:|:---:|
> | Si-GT-wo IntraNetAttn | 88.27 | 73.66 | 87.26 | 70.87 |
> | Si-GT-wo InterNetAttn | 88.22 | 73.40 | 87.29 | 71.06 |
> | Si-GT | 88.32 | 73.67 | 87.39 | 71.82 |
>
> | Ablation Configuration | Segment-$t_\text{width}$ | Segment-$v_\text{max}$ | Sink-$t_\text{width}$ | Sink-$v_\text{max}$ |
> |:---|:---:|:---:|:---:|:---:|
> | Si-GT-wo IntraNetAttn | 97.97 | 97.39 | 97.99 | 97.68 |
> | Si-GT-wo InterNetAttn | 98.12 | 97.83 | 97.98 | 97.44 |
> | Si-GT | 98.36 | 97.78 | 98.53 | 98.63 |
>
> Overall, the full Si-GT model consistently outperforms both ablated variants, demonstrating the effectiveness of modeling both intra-net and inter-net interactions.
>
> ### (3) Ablation of Graphormer’s SP and edge biases.
> In our design, SP captures the global structural context of the interconnect topology (shortest-path hops, accumulated Manhattan distance), while IIN models electrical coupling physics such as resistance accumulation and aggressor–victim interaction. To verify this, we conducted ablations removing SP and distance biases, and the results are as follows:
>
> | Ablation Configuration | Segment-$\hat{D}_\text{vic}$ | Segment-$\hat{D}_\text{agg}$ | Sink-$\hat{D}_\text{vic}$ | Sink-$\hat{D}_\text{agg}$ |
> |:---|:---:|:---:|:---:|:---:|
> | Si-GT-wo $\Phi_\text{sp} \Phi_\text{d}$ | 87.25 | 72.84 | 87.03 | 71.50 |
> | Si-GT-w $\Phi_\text{sp} \Phi_\text{d}$| 88.32 | 73.67 | 87.39 | 71.82 |
>
> | Ablation Configuration | Segment-$t_\text{width}$ | Segment-$v_\text{max}$ | Sink-$t_\text{width}$ | Sink-$v_\text{max}$ |
> |:---|:---:|:---:|:---:|:---:|
> | Si-GT-wo $\Phi_\text{sp} \Phi_\text{d}$ |97.74 | 97.13 | 97.59 | 97.57 |
> | Si-GT-w $\Phi_\text{sp} \Phi_\text{d}$| 98.36 | 97.78 | 98.53 | 98.63 |
>
> The performance gain demonstrates that SP doesn't conflict with IIN attention, and it supplies additional global context that strengthens Si-GT.
> ### Review5: GraphGPS variants with GNN preprocessing.
> GraphGPS includes a built-in message-passing GNN for local aggregation. Adding additional GNN encoder before GraphGPS would break the fairness of the comparison. Additionally, in our preliminary tests, this setup also led to severe overfitting.
>
> ### Review6: Performance gains over added complexity.
> While GraphGPS is a competitive baseline, it requires 422,417 parameters, significantly exceeding Si-GT’s 282,029 parameters. This demonstrates that Si-GT’s performance gains from its domain-informed inductive biases rather than from model size or added complexity.
>
> ### Review7: Presentation issues.
> We have revised Figure 3 and included the ablation setup and experimental details in Appendix D3.

---

### Official Review · Reviewer_nnvC · 2025-11-01

**Soundness:** 3
**Presentation:** 3
**Contribution:** 3
**Rating:** 6
**Confidence:** 2

**Summary:**

The authors addressed the challenge of signal integrity analysis in modern integrated circuits, where crosstalk-induced delay variations and glitches caused by capacitive coupling between interconnects can lead to performance degradation and functional failures. Traditional SPICE simulations, while accurate, are computationally expensive and inefficient for large-scale designs. They proposed Si-GT, a transformer-based framework that enables fast and accurate signal integrity analysis by incorporating virtual NET tokens for net-level representation, mesh pattern encoding for capturing high-order coupling structures, and an IIN attention mechanism to model both intra- and inter-net interactions.

**Strengths:**

* The paper is well written, easy to follow
* The authors performed various experiments with good performances.
* The problem the authors aimed to address is an important one in this field.

**Weaknesses:**

* The authors evaluated the model performance with fixed hyperparameters. However, it would be more informative for other AI researchers if a broader hyperparameter search space were explored and a hyperparameter sensitivity analysis were conducted to examine how each hyperparameter affects the model’s performance.
* In Figure 6, the performance of the graph transformer–based model is not clearly visible. Although this does not prevent readers from understanding the main conclusion of the experiment, there could be a better way to visualize the results.
* It appears that the experiments were conducted only once and the performance was reported based on that single run. To ensure that the model’s performance is not dependent on a specific random seed but is statistically meaningful, it is necessary to repeat the experiments multiple times and report the mean and standard deviation of the performance.

**Questions:**

See the 'weakness' part

---

> ### Author Response · Authors · 2025-11-22
>
> We greatly appreciate the reviewer for the positive comments and feedback. We respond to each comment below and sincerely hope that our responses will properly resolve your concerns.
>
> ## Review1: Broader hyperparameter search and sensitivity analysis.
> A full hyperparameter sweep is computationally prohibitive in our setting, as each training run requires processing 387k SPICE-simulated delay and glitch examples, making an exhaustive grid or random search infeasible. While the main paper reports a single final configuration for clarity, we conducted targeted hyperparameter exploration during development. Specifically, for Si-GT, we varied the embedding dimension, the number of MPE layers, and the number of transformer layers to assess model stability and sensitivity. For the baseline GT architectures, we adopted the recommended configurations from their original papers, and we provide the full details in Appendix C.3. The corresponding parameter counts are summarized in Table 7.
>
> Our experiments show that GraphGPS is a strong baseline. However, it requires 422,417 parameters, substantially more than Si-GT’s 282,029 parameters. This suggests that Si-GT’s performance gains from its domain-informed inductive biases rather than increased model size or complexity.
>
> ## Review2: Visibility of the graph transformer curves in Figure 6.
> We appreciate the reviewer’s observation. We have updated Figure 6 for better contrast.
>
> ## Review3: Running experiments multiple times and reporting mean ± std.
> We agree with the reviewer that multi-seed evaluation strengthens statistical reliability. As noted, as model training is computationally expensive, our initial submission used a fixed seed for each model to keep comparisons fair and feasible. In our experiments, the observed variations are small, typically below 0.01 in mean relative accuracy. Given the minimal fluctuations and for the sake of clarity and brevity, we didn't report the multi-seed standard deviations in the main paper, but we will include an explanation in the appendix.

---

### Author Response · Authors · 2025-11-22

We sincerely thank all reviewers for their valuable feedback and constructive comments. **We have submitted a revised version of the paper, with all updates and modifications highlighted in blue for ease of review.** Below, we summarize the key improvements and changes made in the revision.

**Figure Updates**
- Updated Figure 3 to illustrate the mesh subgraph extraction and the node features derived from the input interconnect graph.
- Updated Figure 6 to improve the contrast of the runtime comparison between SPICE and GT models.

**Revised Text**
- In the Related Works section (ML for SI), we clearly describe the limitations of prior approaches, including routing-free crosstalk prediction and GraphCAD.
- In Section 4.2, we clarify the motivation for introducing spatial encoding and edge encoding as attention bias terms.
- In the Ablation Study section, we added forward references to Appendix D.3 for the ablation experiment setup and additional experiments.
- Added a new subsection in Appendix D.3 detailing the ablation experiments.

**Revised Table**
- Updated Table 5 to include ablation results of intra-net and inter-net attention terms.

Thank you again for your time and thoughtful review. We hope that the revisions and added clarifications fully address your concerns.

---

### Meta-Review · Area_Chair_x1X7 · 2026-01-07

**Summary:**

The paper proposes Si-GT, a Graph Transformer framework designed for efficient Signal Integrity (SI) analysis in Integrated Circuit design, specifically targeting crosstalk-induced delay and glitches. The method introduces three key physics-informed components: virtual <NET> tokens to encode net-level characteristics (slew rate, direction), Mesh Pattern Encoding to capture local capacitive coupling structures, and an Intra-Inter Net attention mechanism to explicitly model signal propagation and coupling interactions. To validate the model, the authors construct and benchmark a large-scale synthetic dataset based on SPICE simulations using Intel 14nm parameters, demonstrating significant speedups over SPICE and accuracy improvements over general-purpose GNNs and Graph Transformers.

**Reviewer Concerns:**

Addressed Concerns:
1. Missing Baselines: Reviewers Eseo and kjJC noted the absence of comparisons to recent domain-specific works like GraphCAD. The authors clarified that these baselines are closed-source and target different tasks, making direct comparison infeasible.
2. Dataset Realism: Reviewers kjJC, Eseo, and DwGg questioned the use of synthetic "2-aggressor + 1-victim" topologies versus real industrial netlists. The authors justified this by explaining that capacitive coupling is physically dominated by immediate neighbors and that pairwise abstraction is consistent with commercial sign-off tool methodologies. Reviewer kjJC explicitly accepted this justification, acknowledging the setup is consistent with commercial tools.
3. Ablation Studies: Reviewer Eseo requested fine-grained ablations to disentangle the contributions of specific attention biases. The authors provided these new experiments in the rebuttal, quantifying the distinct gains from Intra-Net and Inter-Net attention modules.

Outstanding Concerns:
1. Access to Industrial Data: While the synthetic dataset is physically grounded and rigorous, the lack of evaluation on proprietary industrial post-routing netlists remains a limitation noted by Reviewers DwGg and Eseo, though this is a common constraint in the EDA domain due to IP restrictions.

**Reviewer Scores:**

Reviewer kjJC (2 -> 6): The reviewer explicitly stated, "I will raise my ratings for this paper" after being convinced by the authors' justification of the dataset's realism and alignment with commercial tools.

---

### Decision · Program_Chairs · 2026-01-26

Accept (Poster)